# *In vivo* pharmacokinetics of ginsenoside compound K mediated by gut microbiota

**Ming-Si Deng[1,2], Su-tian-zi Huang[3], Ya-Ni Xu[2], Li Shao[4], Zheng-Guang Wang◉[5]\*, Liang-Jian Chen[1]\*, Wei-Hua Huang[3,6]**

1 Department of Stomatology, the Third Xiangya Hospital of Central South University, Central South University, Changsha, China, 2 Department of Orthodontics, Changsha Stomatological Hospital, Hunan University of Chinese Medicine, Changsha, China, 3 Department of Clinical Pharmacology, Xiangya Hospital, Central South University, Changsha, China, 4 Department of Pharmacognosy, School of Pharmacy, Hunan University of Chinese Medicine, Changsha, Hunan, China, 5 Department of Spinal Surgery, the Third Xiangya Hospital of Central South University, Central South University, Changsha, China, 6 FuRong Laboratory, Changsha, Hunan, China

\* wzg19830216@163.com (ZGW); jian007040@sina.com (LJC)

**Data Availability Statement:** All relevant data are within the paper.

**Funding:** This research was supported by the National Natural Scientific Foundation of China

## Abstract

Ginsenoside Compound K (GCK) is the main metabolite of natural protopanaxadiol ginsenosides with diverse pharmacological effects. Gut microbiota contributes to the biotransformation of GCK, while the effect of gut microbiota on the pharmacokinetics of GCK *in vivo* remains unclear. To illustrate the role of gut microbiota in GCK metabolism *in vivo*, a systematic investigation of the pharmacokinetics of GCK in specific pathogen free (SPF) and pseudo-germ-free (pseudo-GF) rats were conducted. Pseudo-GF rats were treated with non-absorbable antibiotics. Liquid chromatography tandem mass spectrometry (LC–MS/MS) was validated for the quantification of GCK in rat plasma. Compared with SPF rats, the plasma concentration of GCK significantly increased after the gut microbiota depleted. The results showed that GCK absorption slowed down, $T_{max}$ delayed by 3.5 h, $AUC_{0-11}$ increased by 1.3 times, $CL_{z/F}$ decreased by 0.6 times in pseudo-GF rats, and $C_{max}$ was 1.6 times higher than that of normal rats. The data indicated that gut microbiota played an important role in the pharmacokinetics of GCK *in vivo*.

## Introduction

Ginsenoside Compound K (GCK), a protopanaxadiol (PPD)-type saponin, is the main metabolite of natural ginsenosides from *Panax* species biotransformed by gut microbiota *in vivo* [1]. Comparing with the parent saponins, recent studies find that GCK shows stronger and broader pharmacological effects such as hepatoprotective [2, 3], anti-atherosclerosis [4, 5], anti-diabetes [6, 7], neuroprotection [8], anti-inflammatory [9, 10] and anti-tumor activities [11]. Currently, GCK tablets are being tested in patients with rheumatoid arthritis (ClinicalTrials.gov Identifier: NCT03755258). However, the fate of GCK pertinent to pharmacokinetics *in vivo* is not well-known.

(82074000, 81903784), the Hunan Provincial Natural Science Foundation of China (2023JJ30459, 2024JJ5585, 2024JJ5529, 2024JJ9533), the Scientific Research Program of Furong laboratory (2023SK2083), the Scientific Research Project of Department of Education of Hunan Province (20K136).

**Competing interests:** The authors have declared that no competing interests exist.

Evidences from several animal and clinical studies suggest that GCK is safe and well-tolerated [12, 13]. However, relative poor bioavailability and great individual variations of pharmacokinetics and pharmacodynamics limit the development and application of GCK [14]. Gene polymorphisms only partially explain the individual pharmacokinetics variation of GCK [15]. A clinical trial demonstrates that the pharmacokinetics of GCK is influenced by diet, but the mechanism remains unclear [14]. Gut microbiota plays an important role in drug metabolism [16]. Some studies have shown that probiotics and prebiotics improve the metabolism of the parent saponins of GCK [17, 18]. Moreover, *in vitro* studies also demonstrate that GCK can be biotransformed by gut microbiota [19]. However, the effects of gut microbiota on the pharmacokinetics of GCK are not well investigated.

In this study, the antibiotics including ampicillin, vancomycin, neomycin sulfate and metronidazole were used to construct a pseudo-germ-free (GF) rat model. A Liquid chromatography tandem mass spectrometry (LC-MS/MS) was validated for the quantification of GCK in rat plasma. Compared with the specific pathogen free (SPF) group, the area under the concentration-time curve (AUC) and the CLz/F of GCK were obviously increased in the pseudo-GF group. Our results implied that gut microbiota played an indispensable role in the metabolism of GCK *in vivo*.

## Materials and methods

### Chemicals and reagents

Ampicillin, vancomycin, neomycin sulfate and metronidazole were purchased from BBI Life Sciences Co., LTD. (Shanghai, China). GCK (purity $\geq$ 98.0%) was purchased from Baoji Hebest Biotechnology Co., LTD. (Shanxi, China). HPLC grade acetonitrile (ACN) was provided by Merck (Darmstadt, Germany).

### Ethics statement

Male Sprague-Dawley (SD) rats (8 weeks old, 180–220 g) were purchased from Hunan SJA Laboratory (Hunan, China) (Certificate Number: SCXK-2019-0004). The rats were reared in a controlled environment with relative humidity (60 ± 5%), temperature (20–24˚C), and maintained light and darkness for 12 hours each day. All animal works were approved by the experimental animal welfare ethical Review Institution of Central South University (Hunan, China. Approved number: CSU-2022-0003).

### Construction of a pseudo-germ-free rat model

SD rats were randomly divided into two groups: normal control group (SPF group, n = 7) and pseudo-GF group (n = 7). To deplete the gut microbiota, a solution of ampicillin (50 mg/kg), vancomycin (25 mg/kg), neomycin sulfate (50 mg/kg), and metronidazole (50 mg/kg) were orally administered. Fresh antibiotic concoction with fed ampicillin in water (0.25 g/L) were prepared every day for a week. The feces were collected three days after withdrawal, and then the gut bacterial DNA was extracted with a stool DNA kit (Omega, D4015-01) and measured by Nonodrop 3000 spectrophotometer (Shimadzu, Japan) [20].

### Instruments and condition

The mass spectrometer analysis was performed on QTRAP 6500$^+$ low mass spectrometers. The Phenomenex LUNA C18 (2) Reversed Phase (Phenomenex, 150 × 2.0 mm, i.d., 5 μm) was used for the chromatographic separation with a mobile phase of acetonitrile (A) and the aqueous phase of 2 mM ammonium acetate (B) at a flow rate of 0.3 mL/min. The gradient elution

procedure was set at 0−2 min: 20%A, 2−3 min: 20−65%A, 3−4 min: 65−75%A, 4−5 min: 75 −90%A, 5−6 min: 90−100%A, and the re-equilibration time was set for 5 mins with 20%A after gradient elution. The tailing factors under the selected chromatographic conditions were calculated as 0.993 and 0.996 for GCK and PPD, respectively, while the theoretical plates were measured as $1.13 \times 10^4$. The injection volume was 10 μL with the temperature of column at 40˚C. The mass spectrometer parameters were optimized as follow, Curtain Gas (CUR), 35; collision gas, 9;, ionspray voltage,-4500 V; temperature, 450˚C; curtain gas, 35 psi; ion source gas 1, 50 psi; ion source gas 2, 55 psi. Both GCK and PPD were analyzed in multiple reaction monitoring (MRM) mode, shown as Fig 1.

## Experimental protocol

After fasted for 12 h, all animals were once orally administrated with a medium dose of GCK (22.5 mg/kg) [13] in 0.9% normal saline. Blood samples were collected from the tail vein at 0 h, 0.5 h, 1 h, 1.5 h, 2.5 h, 3 h, 3.5 h, 4 h, 5 h, 6 h, 7 h, 9.5 h, and 11 h after oral administration. All blood samples were immediately centrifuged at 4000 rpm for 20 min at 4˚C. Approximate 100 μL of supernatant was collected and stored at -80˚C for analysis. Non-compartmental model analysis was operated on DAS 2.0 software of the Chinese Pharmacological Society, and pharmacokinetic parameters were calculated accordingly.

## Blood sample preparation

The blood samples (50 μL) were mixed with 150 μL of digoxin solution (Internal standard (IS), 3.33 ng/mL), subsequently vortex-mixed and centrifuged at $1.4 \times 10^4$ rpm (4˚C) to remove the precipitated proteins for 10 min, respectively. Finally, 100 μL of supernatant was subjected to LC-MS/ MS analysis.

## Calibration standards and quality controls (QC)

The primary stock solutions of GCK (1.38 mg/mL), and PPD (1.14 mg/mL) were prepared by dissolving them in methanol, respectively. The working solution was diluted with methanol. The standard calibration curve and QC samples were prepared by spiking 5 μL of working solution with 45 μL of blank rat plasma. The solution of IS was diluted to 3.33 ng/mL using acetonitrile. Calibration standards and QC were validated according to the 2018 FDA Guidelines.

## Validation of method

**Selectivity.** Each blank plasma sample from six rats spiked with/without IS or IS and analytes were analyzed to assess the selectivity. The peak areas near all the analytes in the blank plasma must be less than 20% of all the analytes, respectively, which were less than 5% of blank plasma spiked with IS and analytes. Additionally, the chromatographic separation between IS and analytes must be achieved without interference on the quantification of all the analytes.

**Linearity and quantification.** The calibration curves were constructed in a weighted (1/ $X^2$) linear least squares regression model, and rectified by analyte/IS peak area ratios versus the standards. The correlation coefficient ($R^2$) of calibration curves were qualified more than 0.99. The low limits of quantification (LLOQ) were defined as the lowest concentration for the calibration curve with signal-to-noise ratios (S/N) about 10 at least. Six LLOQ samples were repeatedly analyzed with the concentration acceptance due to accuracy within 85%–115%.

**Precision and accuracy.** Precision was evaluated through relative standard deviation (RSD), while accuracy was assessed using QC samples. The LQC, MQC, and HQC samples

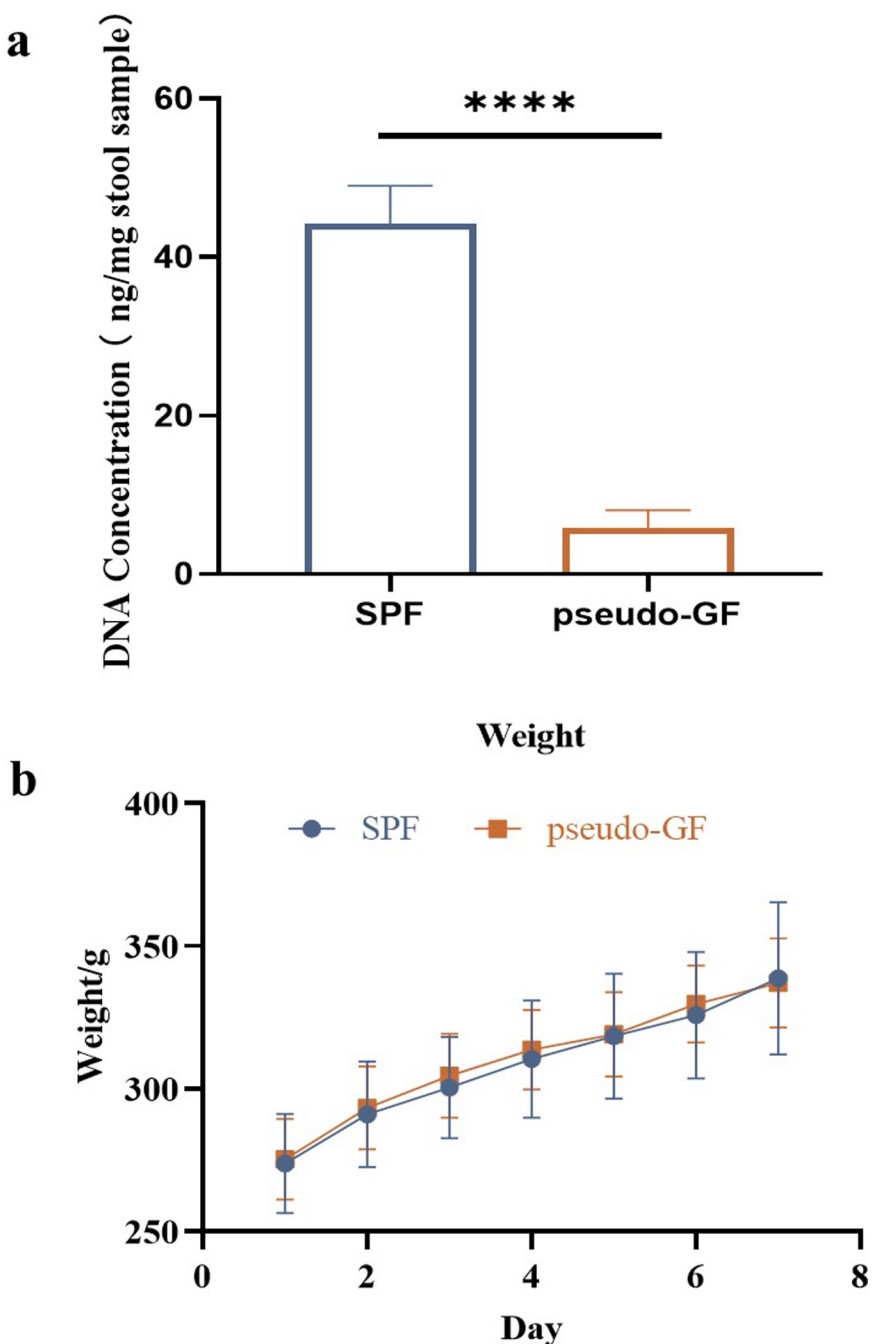

**Fig 1. Validation of pseudo-GF model in rats.** (**a**) Body weights of rats in both normal and pseudo-GF rats after oral antibiotics (n = 5); (**b**) Concentration of gut microbial DNA in feces (data were expressed as mean ± SD, n = 7); SPF, normal control group; pseudo-GF, pseudo-germ-free group, ****$p < 0.0001$.

were quantified to calculate the intra- and inter-day precision and accuracy. The inter-day precision and accuracy were evaluated by determining QC samples in three successive days. Generally, the variability should be within 15%.

**Extraction recovery.** The LQC, MQC, and HQC plasma samples with six replicates at each group were used to evaluate extraction recovery which was assessed by calculating the

ratios between peak areas of QC samples and/with post-extracted blank samples spiked with analytes and IS. The matrix effect was assessed by calculating peak areas ratios of post-extracted samples and/with spiked with IS and analytes. The variability was feasible to be less than 20%.

**Stability.** The LQC, MQC, and HQC samples were stored at –80°C for thirty days. The employed samples were tested to assess long-term stability. Six replicates of all QC samples were frozen at –80°C and thawed at 25°C for three cycles, which were then determined to assess the freeze/thaw cycle stability. The variabilities of precision and accuracy should be within15%.

## Statistical analysis

Pharmacokinetics parameters including AUC, terminal elimination half-life ($T_{1/2}$), mean retention time (MRT), apparent distribution volume (Vz/ F), and clearance rate (CL) were calculated using non-compartmental model analysis (DAS 2.0 software, Chinese Pharmacological Association, China). Other parameters, such as maximum plasma concentration ($C_{max}$) and time to maximum plasma concentration ($T_{max}$), were obtained directly from the plasma concentration-time diagram. All data were expressed as mean ± SD. Significant differences were marked by the symbols ** $p < 0.01$, *** $p < 0.001$, and **** $p < 0.0001$.

## Result

### Pseudo-germ-free rat model construction

After oral administration with antibiotic cocktails, the concentration of microbial DNA significantly decreased in the pseudo-GF rats. Meanwhile, the body weight showed no significant difference between the two groups (Fig 1). Therefore, the results indicated that the gut microbiota was extremely disrupted in the pseudo-GF rats treated with antibiotic cocktails.

### Method validation

**Selectivity.** The analytical method exhibited good selectivity for the detection of GCK in plasma samples. The typical MRM chromatograms of GCK, protopanaxadiol (PPD) and IS in blank rat plasma were shown in Figs 2 and 3. No significant matrix interference was found around the chromatographic retention time of analytes and IS in six different blank plasma samples.

**Linearity.** The linearity range, calibration curves, correlation coefficients and LLOQ of the GCK and PPD in plasma samples were shown in Table 1. Both regression equations showed good linearity ($R^2 \geq 0.99$), and the LLOQ of GCK and PPD were 5.39 and 4.45 ng/mL, respectively. All QC samples were within the standard range (RSD $\leq$ 15%). The developed method was sensitive for analyzing the GCK and PPD in plasma samples.

**Precision and accuracy.** The results of inter- and intra- day precision and accuracy of GCK and PPD were listed in Table 2. The precision of GCK and PPD in QC samples ranged from 3.34% to 9.25%, and 3.73% to 9.09%, respectively. The inter- and intra-day accuracy of GCK and PPD in QC samples were ranged from 97.2% to 104.6%, and 92.2% to 99.6%, respectively. The results indicated that the method was validated with good precision and accuracy.

**Extraction recovery.** The results of extraction recovery were listed in Table 3. The mean of extraction recovery of GCK at three levels of QC samples was 77.80%, 65.22%, and 72.85%, respectively. Deviation at three levels was within 15%. It indicated that a high and reproducible extraction recovery of GCK in this method.

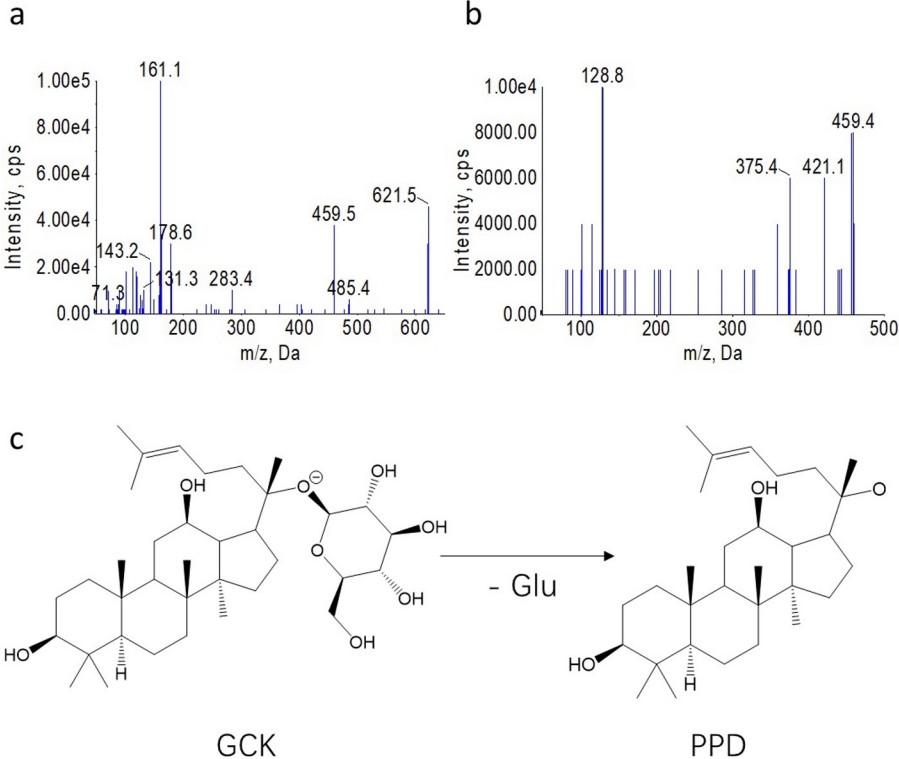

**Fig 2.** Mass spectra and fragmentation pathways of GCK (a), PPD (b) in negative ion mode. (c) The MRM transitions of GCK and PPD.

**Stability.** Comprehensive analysis of freeze/thaw stability and long-term stability, the precision of GCK and PPD in three levels of QC samples were ranged from 2.88%–10.65%, and 3.21%–9.29%, respectively. The accuracy of GCK and PPD in three levels of QC samples were ranged from 95.56%–106.47%, and 89.65%–101.93%, respectively (Table 4). The results suggested that all analytes were stable after storing at –80˚C for 30 days or 3 freeze-thaw cycles.

**Pharmacokinetics of GCK.** To further confirm whether the pharmacokinetics of GCK was changed after that the gut microbiota was disrupted *in vivo*. The chromatograms of GCK were shown in Fig 4. Fig 5 showed the concentration-time curves of GCK in SPF and pseudo-GF rats, and Table 5 summarized the pharmacokinetics parameters. Compared with SPF rats, GCK absorption in pseudo-GF rats were slowed down, $T_{max}$ were delayed by 3.5 h, $AUC_{0-11}$ in pseudo-GF rats were increased by 1.3 times, $CL_{z/F}$ were decreased by 0.6 times (P <0.01), and $C_{max}$ was 1.6 times higher than that of SPF rats. However, the predominant metabolite, PPD, was not detected in the two groups in this study.

## Discussion

The fates of many natural products are related with gut microbiota *in vivo*. The antibiotics alter the characteristics and function of gut microbiota [21]. In order to better understand the impact of gut microbiota on GCK metabolism and its pharmacokinetics, we constructed a pseudo-sterile rat model on which a mixture of antibiotics was selected to disrupt a broad spectrum of both $G^+$ and $G^-$ bacteria [22]. The result of the microbial DNA concentration displayed that the combination of broad-spectrum antibiotics efficiently destroyed the gut microbiota.

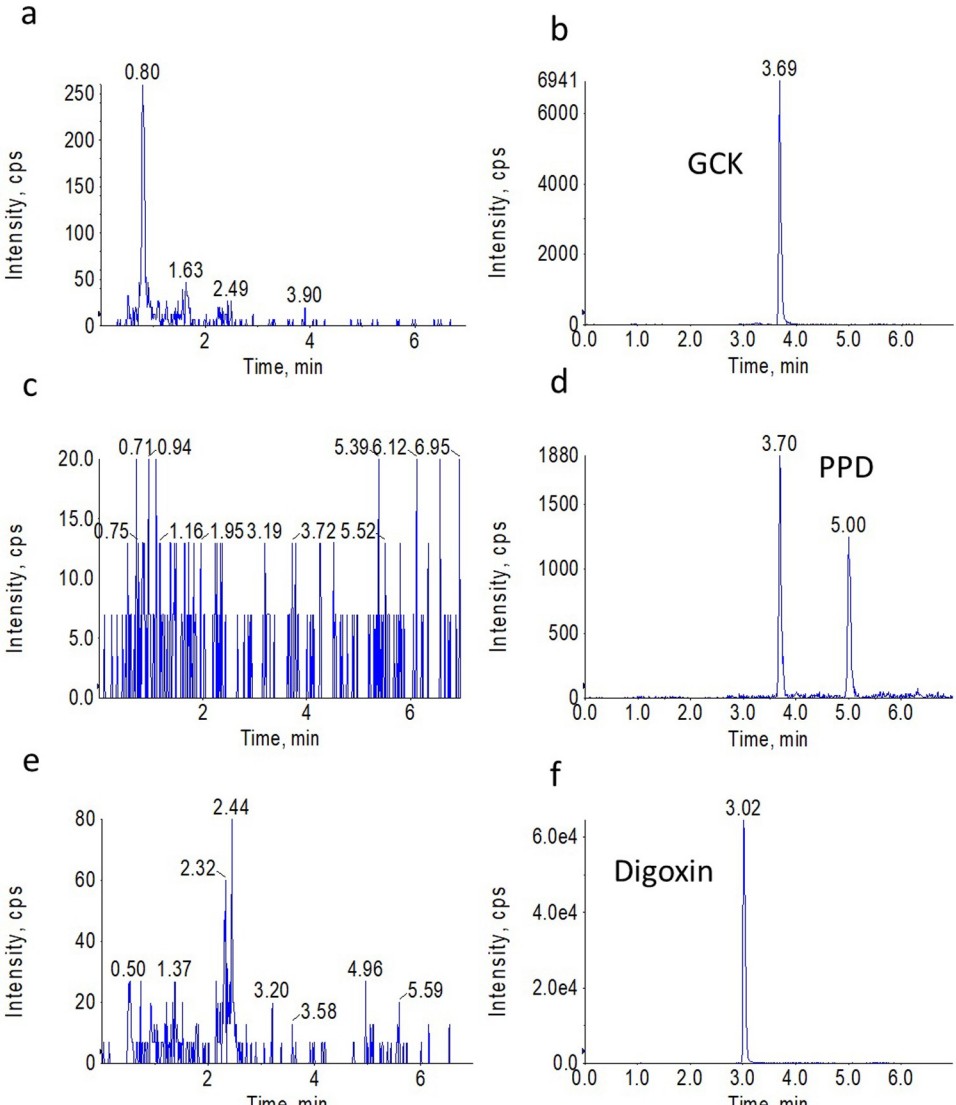

**Fig 3. Typical MRM chromatograms of blank rat plasma and/or spiked with analytes and IS.** (**a**), (**c**), and (**e**) were the chromatograms of blank plasma; (**b**), (**d**), and (**f**) were the chromatograms of GCK, PPD and digoxin, respectively.

To analyze the metabolism of GCK *in vivo*, the different check-point concentrations of the parent drug and PPD in plasma were measured in both groups after a single oral administration. Compared with the SPF group, the concentration of GCK was significantly higher, but the $T_{max}$ was delayed. Meanwhile, the exposed time was longer after oral administration, and the AUC increased by approximately twice in the pseudo-GF group. These results indicated that the pharmacokinetics of GCK greatly changed with the depletion of gut microbiota *in vivo*.

In view of our data, gut microbiota played an important role in the metabolism of GCK *in vivo*. Moreover, when the bacterial community was out-of-balance, the biotransformation of GCK was severely altered due to the catalyzing enzyme secreted by gut microbiota. Besides, the host metabolism of GCK may be affected by the dysbiosis of gut microbiota. Additionally, the data showed that the individual variation of GCK pharmacokinetics was observed in both

**Table 1. Linearity range, correlation coefficients (r), calibration curves and LLOQ of GCK and PPD in plasma samples.**

| Compounds | Linear range (ng/ml) | Correlation coefficients (r) | Calibration curves | LLOQ (ng/ml) |
|---|---|---|---|---|
| GCK | 5.39–690 | 0.9965 | y = 0.000191x+0.00078 | 5.39 |
| PPD | 4.45–570 | 0.9938 | y = 0.000224x+0.000474 | 4.45 |

**Table 2. Precision and accuracy of GCK and PPD.**

| Compounds | Conc. added (ng/mL)[a] | Intra-day (n = 6) | | | Inter-day (n = 18) | |
|---|---|---|---|---|---|---|
| | | Accuracy (%) | Precision (%) | | Accuracy (%) | Precision (%) |
| GCK | 5.39 | 97.59 | 6.17 | | 97.17 | 7.21 |
| | 10.8 | 99.80 | 9.25 | | 102.4 | 7.86 |
| | 86.3 | 104.6 | 3.34 | | 103.6 | 6.65 |
| | 552 | 102.4 | 4.22 | | 98.82 | 7.55 |
| PPD | 4.45 | 96.29 | 8.50 | | 99.56 | 9.09 |
| | 8.91 | 92.18 | 4.16 | | 99.28 | 9.04 |
| | 71.3 | 99.60 | 4.41 | | 95.47 | 6.22 |
| | 456 | 95.61 | 3.73 | | 94.17 | 5.67 |

[a]conc. is the abbreviation of concentration.

**Table 3. Extraction recovery of GCK in rat plasma (mean ± SD[a]).**

| Compounds | Conc. (ng/mL)[b] | Recovery (%) (n = 6) | RSD (%)[c] |
|---|---|---|---|
| GCK | 10.8 | 77.80±11.21 | 14.41 |
| | 86.3 | 65.22±8.25 | 12.65 |
| | 552 | 72.85±10.66 | 14.64 |
| PPD | 8.91 | 77.75±8.44 | 10.85 |
| | 71.3 | 81.61±4.53 | 5.55 |
| | 456 | 87.75±4.20 | 4.78 |

[a] SD: standard deviation.

[b] conc. is the abbreviation of concentration.

[c] RSD, relative standard deviation (%) = standard deviation/ mean.

**Table 4. Freeze/thaw stability and long-term stability of GCK in rat plasma.**

| Compounds | conc. (ng/mL)[a] | Freeze/thaw stability (n = 6) | | long-term stability (n = 6) | |
|---|---|---|---|---|---|
| | | Accuracy (%) | Precision (RSD %) | Accuracy (%) | Precision (RSD %)[b] |
| GCK | 10.8 | 102.76 | 9.07 | 95.56 | 10.65 |
| | 86.3 | 106.47 | 3.37 | 103.32 | 3.57 |
| | 552 | 104.11 | 4.15 | 103.91 | 2.88 |
| PPD | 8.91 | 92.44 | 6.64 | 96.17 | 9.29 |
| | 71.3 | 93.50 | 5.51 | 97.83 | 8.86 |
| | 456 | 89.65 | 3.21 | 101.93 | 4.13 |

[a] conc. is the abbreviation of concentration.

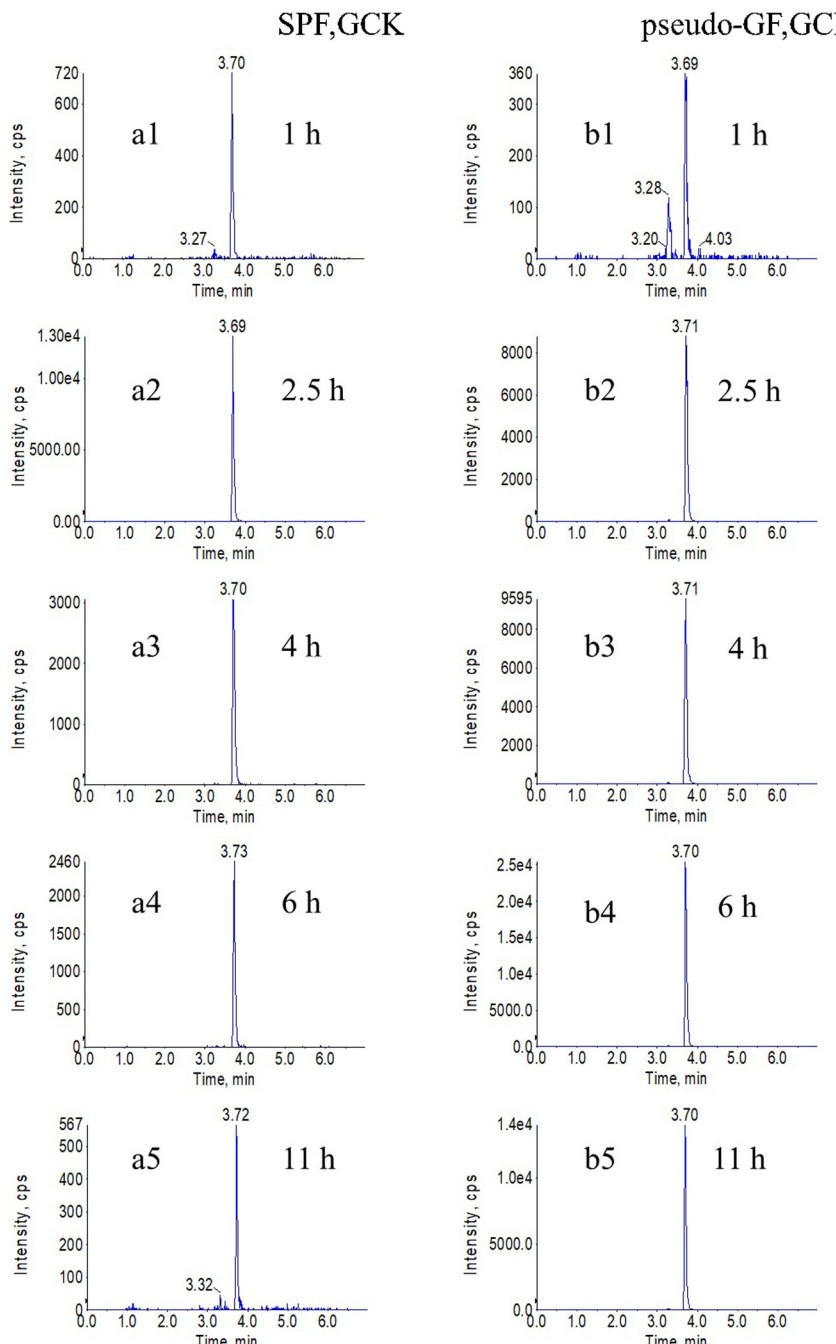

**Fig 4. Typical MRM Chromatograms of GCK at checkpoints in the plasma of SPF group and pseudo-GF group.** (**a1**), (**a2**), (**a3**), (**a4**), and (**a5**) were typical MRM chromatograms of GCK at 1 h, 2.5 h, 4 h, 6 h, and 11 h in SPF group, respectively. (**b1**), (**b2**), (**b3**), (**b4**), and (**b5**) were typical MRM chromatograms of GCK at 1 h, 2.5 h, 4 h, 6 h, 11 h in pseudo-GF group, respectively.

groups. Herein, the diversity of gut microbiota may lead to significant variations in the metabolic behavior of GCK.

In a clinical trial, the $T_{1/2}$ of PPD in the human plasma sample reaches more than 19 hours, while the $T_{max}$ and $C_{max}$ of PPD are approximate 25 hours and 4.2 ng/mL after a single dose of

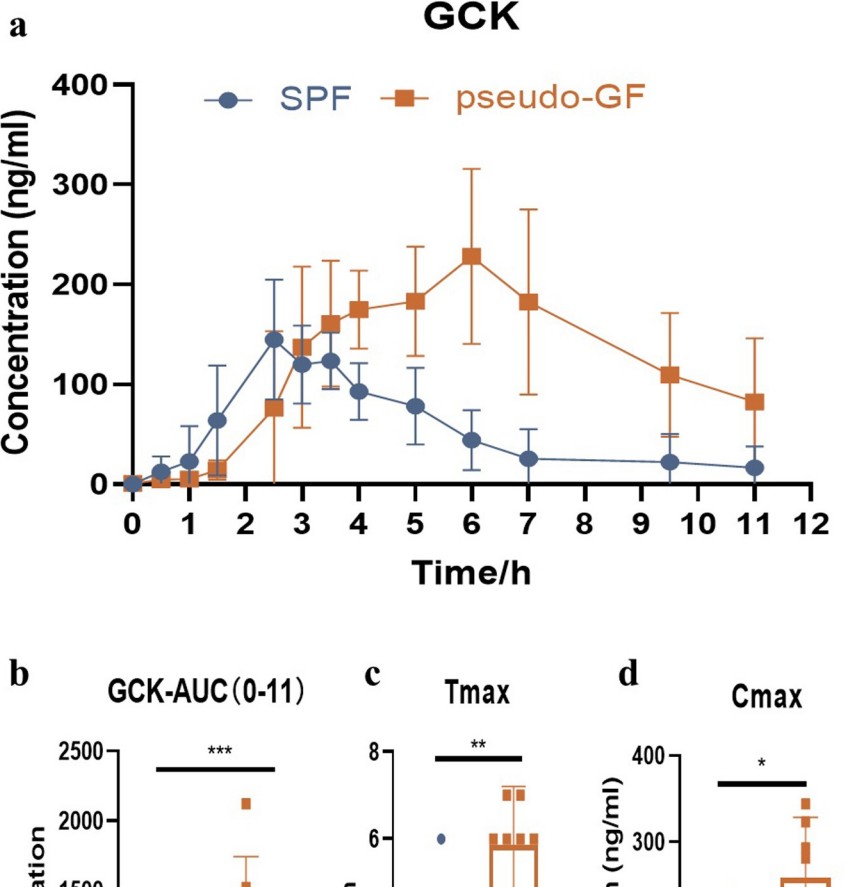

**Fig 5. The mean plasma concentration–time profiles of GCK in rat plasma after intragastric administration of GCK (22.5 mg/kg); each point and bar represent the mean ± SD (n = 7), \*$p < 0.05$, \*\*$p < 0.01$ and \*\*\*$p < 0.001$.**

oral GCK [14]. Our method was more sensitive than the employed methods for the detection of GCK and PPD in plasma samples. However, PPD was not detected in our study due to the possibility of poor generation and absorption of PPD after oral administration of GCK in rats. Therefore, the pharmacokinetics parameters of PPD were unable to be calculated for evaluating its pharmacokinetics in this study. Nevertheless, the validated method was greatly applied for the pharmacokinetics of GCK *in vivo*.

## Conclusion

The pharmacokinetics of GCK was significantly altered along with the depletion of gut microbiota in rats. Though GCK is able to be metabolized to generate PPD, which was not detected in rat plasma from both normal and pseudo-GF groups after single oral administration of

**Table 5. Summary pharmacokinetic parameters of ginsenoside CK.**

| Parameters | SPF group | pseudo-GF group | P value |
|---|---|---|---|
| $C_{max}$ (ng/L) | 162.21±46.93 | 258.57±69.97 | < 0.05 |
| $T_{max}$ (h) | 3.36±1.25 | 5.86±1.35 | < 0.01 |
| $T_{1/2}$ (h) | 2.20±1.69 | 3.93±1.65 | 0.077 |
| $MRT_{(0-11)}$ (h) | 4.16±1.11 | 6.05±0.42 | < 0.01 |
| $AUC_{(0-\infty)}$ (ng/L*h) | 678.95±284.40 | 2299.05±1626.69 | < 0.05 |
| $AUC_{(0-11)}$ (ng/L*h) | 586.32±161.33 | 1340.82±401.50 | 0.001 |
| $CL_{z/F}$ (L/h/kg) | 37941.58±13973.39 | 13561.93±6948.47 | < 0.01 |
| $V_{z/F}$ (L/kg) | 96540.94±44906.34 | 63166.94±15378.70 | 0.088 |

$C_{max}$: maximum plasma concentration; $T_{max}$: time of maximum plasma concentration; $T_{1/2}$: half-life of elimination; MRT: mean residence time; AUC: area under the plasma concentration vs time curve; $CL_{z/F}$: clearance rate; $V_{z/F}$: apparent volume of distribution.

GCK. The pharmacokinetics of GCK presented individual variation due to potential binary metabolism of host and gut microbiota.

## Acknowledgments

The authors would like to great acknowledge Mr. Wang Yicheng and Dr. Gao Yongchao of the Institute of Clinical Pharmacology, Central South University, for their technical assistance in technical methods and statistical analysis.

## Author Contributions

**Conceptualization:** Liang-Jian Chen.

**Data curation:** Ming-Si Deng, Su-tian-zi Huang, Ya-Ni Xu, Li Shao, Liang-Jian Chen.

**Formal analysis:** Ming-Si Deng, Su-tian-zi Huang, Ya-Ni Xu.

**Funding acquisition:** Wei-Hua Huang.

**Investigation:** Ming-Si Deng, Su-tian-zi Huang, Zheng-Guang Wang, Wei-Hua Huang.

**Methodology:** Ming-Si Deng, Su-tian-zi Huang, Li Shao, Liang-Jian Chen.

**Project administration:** Ming-Si Deng, Zheng-Guang Wang.

**Validation:** Su-tian-zi Huang, Li Shao, Zheng-Guang Wang, Liang-Jian Chen.

**Visualization:** Li Shao, Zheng-Guang Wang, Liang-Jian Chen.

**Writing – original draft:** Ming-Si Deng.

**Writing – review & editing:** Ya-Ni Xu, Zheng-Guang Wang, Wei-Hua Huang.

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
