## [Decision Letter · Decision Letter 0]

4 Apr 2024

PONE-D-23-42394In vivo pharmacokinetics of ginsenoside compound K mediated by gut microbiotaPLOS ONE

Dear Dr. wang,

Thank you for submitting your manuscript to PLOS ONE. After careful consideration, we feel that it has merit but does not fully meet PLOS ONE’s publication criteria as it currently stands. Therefore, we invite you to submit a revised version of the manuscript that addresses the points raised during the review process.

We look forward to receiving your revised manuscript.

Kind regards,

Muhammad Hanif

Academic Editor

PLOS ONE

Journal Requirements:

2. To comply with PLOS ONE submissions requirements, in your Methods section, please provide additional information regarding the experiments involving animals and ensure you have included details on (a) methods of sacrifice, (b) methods of anesthesia and/or analgesia, and (3) efforts to alleviate suffering.

This research was supported by the National Natural Scientific Foundation of China (82074000, 81903784), the Hunan Provincial Natural Science Foundation of China (2023JJ4971), the Scientific Research Program of Furong laboratory (2023SK2083), the Scientific Research Project of Department of Education of Hunan Province (20K136).

Additional Editor Comments:

mentioned as above

Reviewers' comments:

Reviewer's Responses to Questions

**Comments to the Author**

1. Is the manuscript technically sound, and do the data support the conclusions?

Reviewer #1: Partly

2. Has the statistical analysis been performed appropriately and rigorously? 

Reviewer #1: Yes

3. Have the authors made all data underlying the findings in their manuscript fully available?

Reviewer #1: Yes

4. Is the manuscript presented in an intelligible fashion and written in standard English?

Reviewer #1: Yes

5. Review Comments to the Author

**Reviewer #1:** The authors have presented a pharmacokinetic study of Ginsenoside Compound K (GCK) in rats and the effect of microbiota on the pharmacokinetics of said compound.

The manuscript needs major revision before further consideration for publication. Key information is missing without which the study has no credibility. Occasionally, the methodology is weak and does not fully justify the presented results.

1. In the section ‘Instruments and condition’ MRM transitions of drug molecules were not mentioned. No quantification can be done without this information.

2. Collision gas and its flow rate not mentioned.

3. No re-equilibration time has been provided to the system which is essential for a gradient elution program.

4. Calibration standards and quality control: analytical range not mentioned for QCs and calibration standards.

5. System suitability parameters (tailing factor, theoretical plates) have not been mentioned.

6. The chromatograms are not legible. No information about Molecular ion peaks or daughter ions has been provided.

7. English usage is not up to the standard. Phrases like ‘all animals were intra-gastric administration’ and ‘The peak areas near all the analytes in the blank plasma were satisfied less than 20% of all the analytes’ are few examples.

6. PLOS authors have the option to publish the peer review history of their article (what does this mean?). If published, this will include your full peer review and any attached files.

Reviewer #1: No

---

## [Author Response · Author response to Decision Letter 0]

7 May 2024

Responses to Editor’s and Reviewers’ Comments

Dear Editor-in-chief,

We are submitting a revised manuscript entitled “In vivo pharmacokinetics of ginsenoside compound K mediated by gut microbiota” by Deng et al. for publication as a full paper in the PLOS ONE. We have carefully considered your comments in preparing the revised manuscript. We would like to give a point-by-point response to the concerns. The main manuscript has been revised accordingly. Here, we have enclosed the replies to the reviewers’ comments,

Reviewer’s comments,

1. In the section ‘Instruments and condition’ MRM transitions of drug molecules were not mentioned. No quantification can be done without this information. 

R: Thanks for your comments. The MRM transitions of drug molecules were supplemented in Figure 2.

2. Collision gas and its flow rate not mentioned.

R: Thanks for your comments. The flow rate is 0.3 mL/min; collision gas is 9. We have supplemented this information in the revised manuscript.

3. No re-equilibration time has been provided to the system which is essential for a gradient elution program.

R: Thanks for your comments. The re-equilibration time was set as 5 min in this study. We have supplemented it in the revised manuscript.

4. Calibration standards and quality control: analytical range not mentioned for QCs and calibration standards.

R: Thanks for your comments. Calibration standards and QC were validated according to the 2018 FDA Guidelines.

5. System suitability parameters (tailing factor, theoretical plates) have not been mentioned.

R: Thanks for your comments. The tailing factors under the selected chromatographic conditions were calculated as 0.993 and 0.996 for GCK and PPD, respectively, while the theoretical plates were measured as 1.13 × 104. We have calculated the chromatographic parameters, which have been added in the manuscript.

6. The chromatograms are not legible. No information about Molecular ion peaks or daughter ions has been provided.

R: Thanks for your comments, we have re-provide clearer chromatograms.

7. English usage is not up to the standard. Phrases like ‘all animals were intra-gastric administration’ and ‘The peak areas near all the analytes in the blank plasma were satisfied less than 20% of all the analytes’ are few examples.

R: Thanks for your comments, we have corrected and polished the language.

I hope you will find our corrections worthy of publication in the PLOS ONE.

Best regards.

Yours Sincerely,

Zheng-Guang Wang, PhD

Department of Spinal Surgery,

The third Xiangya Hospital, Central South University

---

## [Editor Report · Decision Letter 1]

3 Jul 2024

In vivo pharmacokinetics of ginsenoside compound K mediated by gut microbiota

PONE-D-23-42394R1

Dear Dr. wang,

We’re pleased to inform you that your manuscript has been judged scientifically suitable for publication and will be formally accepted for publication once it meets all outstanding technical requirements.

Kind regards,

Muhammad Hanif

Academic Editor

PLOS ONE

Additional Editor Comments (optional):

I have gone throuigh the revision submitted by the authors and found satisfactory for the publication
---

## [Editor Report · Acceptance letter]

15 Jul 2024

PONE-D-23-42394R1 

PLOS ONE

Dear Dr. Wang, 

I'm pleased to inform you that your manuscript has been deemed suitable for publication in PLOS ONE. Congratulations! Your manuscript is now being handed over to our production team.

Kind regards, 

on behalf of

Dr. Muhammad Hanif 

Academic Editor

PLOS ONE